# How Psychological Variables Maybe Correlated with Willingness to Get COVID-19 Vaccine: A Nationwide Cross-Sectional Study of Polish Novice Nurses

**DOI:** 10.3390/ijerph192315787

**Published:** 2022-11-27

**Authors:** Joanna Gotlib, Mariusz Jaworski, Ilona Cieślak, Tomasz Sobierajski, Dominik Wawrzuta, Piotr Małkowski, Beata Dobrowolska, Danuta Dyk, Aleksandra Gaworska-Krzemińska, Elżbieta Grochans, Maria Kózka, Jolanta Lewko, Izabella Uchmanowicz, Mariusz Panczyk

**Affiliations:** 1Department of Education and Research of Health Sciences, Faculty of Health Sciences, Medical University of Warsaw, 02-091 Warsaw, Poland; 2Faculty of Applied Social Sciences and Resocialization, University of Warsaw, 00-503 Warszawa, Poland; 3Department of Surgical Nursing, Transplantation Nursing and Extracorporeal Therapies, Medical University of Warsaw, 02-007 Warsaw, Poland; 4Department of Management in Nursing, Medical University of Lublin, 20-081 Lublin, Poland; 5Department of Anaesthesiology and Intensive Care Nursing, Poznan University of Medical Sciences, 60-179 Poznan, Poland; 6Department of Nursing Management, Medical University of Gdansk, 80-211 Gdansk, Poland; 7Department of Nursing, Pomeranian Medical University in Szczecin, 70-204 Szczecin, Poland; 8Department of Clinical Nursing, Faculty of Health Sciences, Jagiellonian University Collegium Medicum in Krakow, 31-501 Krakow, Poland; 9Department of Primary Health Care, Faculty of Health Sciences, Medical University in Bialystok, 15-054 Bialystok, Poland; 10Faculty of Health Sciences, Wroclaw Medical University, 50-367 Wroclaw, Poland

**Keywords:** COVID-19 vaccine, novice nurses, anxiety, self-efficacy, resilient coping, Generalized Anxiety Disorder 7-Item Scale, General Self-Efficacy Scale, Brief Resilient Coping Scale

## Abstract

Introduction: Nurses became the largest medical group exposed to direct contact with the SARS-CoV-2 virus. In this study, we aimed to assess the readiness and motivation for vaccination, as well as the use of sources of information and attitudes toward vaccination depending on the psychological profile. Material and methods: A cross-sectional online survey study was conducted. The study included 145 novice nurses from 8 medical universities who completed 3-year undergraduate studies. Women constituted 97.2% of the respondents (N = 141). The Generalized Anxiety Disorder 7-Item Scale, General Self-Efficacy Scale, Brief Resilient Coping Scale, and an original questionnaire were used. Variables were analyzed with descriptive statistics methods. A *p*-value of <0.05 was considered statistically significant. Results: Among the participants, 73.1% had already been vaccinated against COVID-19 (N = 106). The participants were divided into two groups: G1 (N = 98), characterized by a lower level of anxiety with higher self-efficacy and resilient coping, and G2 (N = 47), with a higher level of anxiety with poorer self-efficacy and resilient coping. The analysis of the potential correlation of psychological pattern with the decision to vaccinate was not statistically significant (*p* = 0.166). Conclusion: Psychological variables may be correlating with motivation, attitudes toward vaccination, and the choice of reliable sources of information about vaccination. Our study demonstrates the key role of two psychological variables, self-efficacy and resilient coping, in this context.

## 1. Introduction

The outbreak of the pandemic, announced by the World Health Organization (WHO) on 11 March 2020, highlighted the current deficiencies in the organization of health care systems in many countries worldwide. However, the problem of the lack of medical staff—in particular, the lack of an adequate number of nurses—became the most visible [1,2]. It is estimated that almost 30 million people worldwide work in nursing, which makes nurses the largest professional group in the global health sector, accounting for 59% of all medical professionals [1].

The problem of the lack of properly trained staff is exacerbated by a noticeable trend in many countries of graduates completing their nursing studies, but leaving the profession after several years of practice [2]. With reference to Benner’s novice to expert theory, the term “novice nurses” describes nursing students/graduates who are just starting their professional work and have no clinical experience, or their length of service is shorter than 2 years [3]. Despite the fact that the number of nurses in the world increased by 4.7 million between 2013 and 2018, numerous countries still face a lack of nursing staff [1]. The global shortage of nurses is estimated to affect about 5.9 million people, with 89% concentrated in low- and middle-income countries [1]. According to OECD data, the average number of nurses per 1000 inhabitants is 8.83 [4]. Poland is one of the countries in Europe with the lowest number of nurses. According to data for 2019, the Polish health care system employed 193,132 nurses out of 299,629 who had the right to practice the profession [5], translating to a rate of 5.10 nurses per 1000 inhabitants [4]. The COVID-19 pandemic triggered a need for organizational changes in health care systems around the world. Nurses were often required to shift to wards designated for patients infected with the new type of coronavirus. Nurses also became the largest medical group exposed to direct contact with the SARS-CoV-2 virus. It should be highlighted that some nurses, referred to as “novice nurses”, began their clinical work in the pandemic; therefore, they had no prior experience of the work environment compared to nurses with longer work experience. This may have exacerbated stress and anxiety as well as affected adaptation to the new situation. Especially that research shows that novice nurses present anxiety behaviors related to such factors as working in a dangerous environment, caring for patients in serious condition requiring comprehensive care, lacking knowledge and experience, and needing to quickly learn new skills not previously used in the workplace, e.g., how to operate medical devices [6]. 

Numerous publications have tackled the issue of study-to-work transitioning during the COVID-19 pandemic [7,8,9,10,11]. During the pandemic, many European countries had to rely on final-year nursing and medical students working side-by-side with health professionals in order to minimize system overload. However, these studies mainly presented the results of qualitative research conducted in small groups of final-year students in some regions of Spain. In March 2020, during the peak of the first wave of the pandemic, final-year nursing students were deployed to work at hospitals before they completed their formal studies and received their diplomas. This was a way of mitigating the risks, both to patients and to the stability of the health care system, due to not having enough nursing staff [12,13,14,15]. For example, early in the year 2020, the understaffing in the Spanish region of La Rioja was so high that the nursing students were offered paid jobs just two weeks after the suspension of regular clinical placements. Their starting salary was equal to that of a newly qualified nurse and they were given positions across a wide range of healthcare settings, from hospital units to care homes or repurposed hotels. In such circumstances, though supervised by registered nurses, the nursing students were treated as full members of the team [16].

Studying motivation toward vaccination among young nurses is important not only for their protection (as our survey showed, the vast majority were vaccinated), but mainly for the sake of patients. Nurses spend a lot of time with patients, which has a real impact on their decision to vaccination [17].

A meta-analysis comparing levels of anxiety and stress and symptoms of depression in a group of novice nurses during the MERS, SARS, and COVID-19 epidemics indicated that the level of anxiety, as measured by the GAD-7 scale, was 30% higher during the COVID-19 pandemic compared to the other two epidemics [18]. Such high anxiety affects the process of adapting to new, stressful situations and intensifies fear for one’s health and life. The pressure and difficulty of adjusting to both a new role and the epidemic crisis created a heavy workload. The pandemic contributed to the development of new factors that might have affected the level of anxiety in newly employed nurses, e.g., the lack of appropriate protective equipment; the high risk of getting infected or infecting loved ones, co-workers, or patients; and the lack of experience and ability to cope with a crisis situation [19]. 

Ensuring adequate protection from the virus, may, on the one hand, decrease the risk of infection associated with contact with a number of patients, and on the other hand, it may help to reduce some of the anxiety in novice nurses. They needed to be protected against the virus, with one of the protective measures being free vaccination. protective measures being free vaccination. Vaccination has become even more important in suppressing the spread of the COVID-19 pandemic, yet these are cognitive, emotional and social processes that shape the public’s willingness to vaccinate. Vaccination raises many emotions, questions and concerns regarding, among other things, its effectiveness and impact on health [20]

COVID-19 vaccine acceptance may also be affected by the types of information sources novice nurses use. Sources of information that may significantly influence COVID-19 vaccine hesitancy include mass media, social media or recommendations from family or friends [21,22]. On the other hand, searching for vaccination information on government websites or obtaining it from a doctor contribute to higher COVID-19 vaccine acceptance [23].

Based on the above data, it is important to pay attention to the level of willingness to receive COVID-19 vaccine in nurses. It should not be assumed that all nurses will have the same high level of willingness for vaccination. Psychological variables are thought to be important for the willingness to be vaccinated against COVID-19 [20] In the case of novice nurses, it is important to take into account psychological variables that will also have an impact on the effective process of adaptation to new conditions.

The psychological variable of resilient coping may be particularly important in the context of adapting to new working conditions. It is a variable that defines human behavior in the context of events that are a source of stress and disturb everyday functioning [24]. A high level of resiliency allows for more effective coping with difficult life situations, which undoubtedly includes the COVID-19 pandemic. In other words, nurses with a high level of resilient coping should be able to cope more effectively with stress and anxiety associated with the COVID-19 pandemic than nurses with a low level. Furthermore, a high degree of resilience should be a motivating factor for nurses to get vaccinated. 

In light of the above-mentioned changing working conditions and external factors that nurses have no control over, the sense of self-efficacy may have a positive impact on the process of adaptation to new working conditions. It is the conviction that one is capable of accomplishing a certain action or goal in spite of various difficulties [25]. Therefore, nurses with a strong sense of self-efficacy will be better able to cope with stressful situations related to changing patient-care conditions. In this context, this psychological variable should also be positively correlating with their willingness to get a COVID-19 vaccine.

In addition to analyzing anxiety in novice nurses, attention should also be paid to nurses’ willingness to be vaccinated, the types of information sources they use to expand their knowledge about the pandemic, and their perception of the vaccine and vaccination. Psychological traits such as sense of self-efficacy and resilient coping may also be important in this context.

This study aimed to assess the readiness and motivation for vaccination, as well as the use of sources of information and attitudes toward vaccination depending on the psychological profile outlined by three variables: anxiety, sense of effectiveness, and resilient coping. With regard to the aim of the study, the following theoretical assumptions were made:I.Novice nurses will have different levels of anxiety, which may be further exacerbated by the situational factor of the pandemic. We will assume that there will be nurses with high, moderate and low levels of anxiety. However, in further analysis, nurses will be divided into a group with high and low anxiety intensity. Taking into account the differences between these two extreme groups will allow for a better understanding of the analyzed phenomenon.II.Psychological traits such as sense of self-efficacy and resilient coping are relatively constant and should not depend on situational factors such as the pandemic. Moreover, the listed variables are correlated with each other. Therefore, they will interact.III.Different levels of psychological traits such as sense of self-efficacy and resilient coping will differentiate novice nurses in terms of their willingness to be vaccinated, use of information sources regarding the pandemic and vaccination, as well as perception of the vaccine and vaccination.

Subsequently, based on the presented assumptions, two hypotheses were formulated and subjected to empirical validation:

**H(1).** 
*The participants characterized by a lower level of anxiety with higher self-efficacy and resilient coping and participants with a higher level of anxiety with poorer self-efficacy and resilient coping will differ in terms of readiness to vaccinate.*


**H(2).** 
*The participants characterized by a lower level of anxiety with higher self-efficacy and resilient coping and participants with a higher level of anxiety with poorer self-efficacy and resilient coping will differ in their use of sources of information about vaccination and the pandemic and perception of the vaccine and vaccination.*


The results on the occurrence of anxiety in the novice nurses group in Poland presented in this paper are the first and, so far, the only quantitative data based on research conducted in a group of graduates who completed nursing studies during the COVID-19 pandemic (June–September 2020) and began their professional work at the peak of the third wave of infections (September–October 2020).

## 2. Materials and Methods

### 2.1. Design and Setting

A cross-sectional online survey study was conducted between July and September 2021. Novice nurses from 8 medical universities offering undergraduate studies in Poland were invited to participate in the study. 

### 2.2. Sampling

Judgment sampling was used in this study. This type of purposive sampling involves units selected for inclusion in a study based on the professional judgment of the researcher. It stands in contrast to probability sampling, in which units are drawn with some probability (e.g., randomly) from the population of interest. 

### 2.3. Local Contects

In Poland, nursing students who complete the 3-year undergraduate nursing study program receive a Bachelor of Nursing degree, and after passing the final university examination they are granted the right to practice the profession by the appropriate Branch of Nurses’ Chamber. In practice, this means that at the same time they successfully complete their first-cycle studies, according to Benner’s theory, they become novice nurses and may start professional practice in Poland or another EU country [26].

### 2.4. Participants

Novice nurses who completed the 3-year undergraduate nursing study program were eligible for the study. In addition to completing the program, entering the nursing profession was the main inclusion criterion. The data package was received from 211 graduates; however, only 145 graduates met the inclusion criteria. The reasons for exclusion from the study were as follows: did not enter the profession, had previous experience in another medical profession, was an unregistered nurse, or did not consent to be contacted after completing the university nursing program, or there was a lack of complete data. The reasons for exclusion were recorded by the coordinators. 

### 2.5. Instruments

The study tool consisted of four sections: (1) demographics, (2) motivations toward COVID-19 vaccines (e.g., Which of the reasons AND to what extent decided that you got vaccinated against COVID-19), (3) vaccine information sources (e.g., Indicate from which sources AND how often you obtain your knowledge about vaccination against COVID-19), and (4) perception of the reasons for non-vaccination (Which of the reasons listed and to what extent do you think contribute to the fact that nearly half of Poles do not want to be vaccinated against COVID-19?). Survey development was informed by the extant literature on COVID-19, vaccines against COVID-19, and vaccine readiness and hesitation among medical staff and medical and nursing students. The questionnaire was developed by means of a brainstorming technique on the part of the authors and a competent judge method. The questionnaire used closed-ended, semi-open, and open-ended questions with Likert-scale responses [27].

The questionnaire also assessed anxiety levels with the use of the Generalized Anxiety Disorder 7-Item Scale (GAD-7) [28,29]. It is based on a four-point Likert scale and is used to assess anxiety and the risk of generalized anxiety disorder (GAD). The questionnaire assessed the ease of occurrence and intensity of the following features: anxiety, tension, nervousness, as well as the ability to control them and difficulty relaxing. The respondents awarded 0 to 3 points, depending on the frequency of occurrence of these features in the 14 days preceding the study (0, not at all; 1, every few days; 2, more frequently than half the days; 3, almost every day). The thresholds of 5, 10, and 15 points marked the occurrence of mild, moderate, and severe anxiety, respectively. A score of 10 points or more was the determinant of a high likelihood of generalized anxiety disorder occurring [30]. The GAD-7 was used with a suggested cut-off point of 10 points to define moderate anxiety and 15 points to define severe anxiety. This questionnaire has been widely used and was reported to have high internal consistency and good test–retest reliability among adults [28,29], adolescents [31,32], and college students [33].

In order to assess the sense of self-efficacy in the novice nurses group, the General Self-Efficacy Scale (GSES) was used [34]. The GSES is a 10-question tool designed to examine an individual’s overall belief in the effectiveness of their coping skills when dealing with emerging difficulties and obstacles. The test was designed for use with adults. Respondents react to given statements on a 4-point scale (1 point for no, 4 points for yes). The total score (10–40) determines the general indicator of self-efficacy, which is then converted into standardized units (sten scores). When interpreting the GSES score, the subjects can be divided into 3 groups: high, average, and low self-efficacy. A score of 30 points or above indicates a high sense of self-efficacy, and this corresponds to a sten score of 7–10; between 25 and 29 points indicates average self-efficacy (sten score of 5–6); and 24 points and lower indicates low self-efficacy (sten score of 1–4). The internal consistency of the GSES was estimated based on a study of 174 people aged 20–55 years. The Cronbach’s alpha coefficient was 0.85. The reliability of the scale, assessed by the test–retest method, was 0.78 after 5 weeks [34].

The Polish version of the Brief Resilient Coping Scale (BRCS) [24], developed by Sinclair and Wallston [35], was used to assess levels of resilient coping among groups of healthy and non-healthy adults, understood in terms of process. The scale includes 4 statements related to various difficult life situations. The selected responses are used to determine the level of resilient coping. BRCS is a short self-assessment tool that takes less than 5 min to complete. Therefore, it can be used for both individuals and groups. The questionnaire instructions explain that responses are based on a Likert scale, ranging from 1 (“definitely does not describe me”) to 5 (“definitely describes me”). Due to the one-factor structure of the scale, calculating the total score consists of adding up the scores given by the respondent. The higher the result, the higher the level of resilient coping. This score can also be represented on a sten scale, with sten scores of 1 to 4 indicating low resilience, 5 to 6 average resilience, and 7 to 10 highly resilient coping. Scale reliability was estimated using Cronbach’s alpha coefficient at 0.625. Because the scale consists of only 4 statements, the obtained reliability indicators can be considered satisfactory. The absolute stability, measured by a 6–7-week interval test–retest in a group of 66 men and women (M age = 40.44 years, SD = 11.78, range 18–68 years), was 0.584 [35].

### 2.6. Data Collection

The limited possibility of in-person contact with the respondents due to the COVID-19 pandemic restrictions imposed by the Minister of Health resulted in the decision to distribute the questionnaire using the LimeSurvey web platform. Coordinators from the participating medical universities shared the link to the online survey to ensure the safety of the study participants [19,36].

### 2.7. Data Analysis

Continuous and categorical variables were analyzed with descriptive statistics methods. The central tendency (mean, M) and dispersion (standard deviation, SD) were determined for continuous variables, while the number (N) and frequency (%) were determined for categorical variables. 

During the first stage, data clustering with the k-means method was performed to distinguish groups of novice nurses based on their psychological patterns [37]. The analysis included selecting and distinguishing groups of similar objects into two clusters. A non-hierarchical clustering algorithm was implemented based on the values calculated for three indices: anxiety, self-efficacy, and resilient coping. Two groups of novice nurses were distinguished: G1 (low level of anxiety with high self-efficacy and resilient coping) and G2 (high level of anxiety with poor self-efficacy and resilient coping). 

Cross tables and Fisher’s exact test or Pearson’s chi-squared test were used to assess the impact of each group on selected categorical variables. Student’s *t*-test was used to assess the difference between groups and selected continuous variables. Cohen’s *d* was determined as a measurement of the effect of differences between means. The following criteria were assumed to assess the measured effect size: very strong ≥0.80, strong 0.50–0.79, average 0.49–0.20, and poor <0.2 [38].

All calculations were performed with Statistica^TM^ 13.3 software (TIBCO Software, Palo Alto, CA, USA). A *p*-value of <0.05 was considered statistically significant for all analyses.

## 3. Results

### 3.1. Sample Characteristics

A total of 145 Polish novice nurses (PNNs) participated in the study. The largest group of participants consisted of those working in hospitals without COVID-19 units (N = 99, 68.3%). Women constituted the majority of respondents (N = 141, 97.2%). The mean age of study participants was 23.3 years (SD = 0.92). The sample group was homogeneous by ethnicity and age. Selected characteristics of the study group are presented in Table 1.

### 3.2. Psychological Patterns

#### 3.2.1. Anxiety

The mean GAD-7 score was 6.66 (SD = 4.96), with a minimum value of 0.0 and a maximum value of 18.0 and it was characterized by moderate right-sided skewness (skew = 0.64), which a slightly larger number of people who obtained lower scores.

#### 3.2.2. Self-Efficacy

The mean GSES score was 26.86 (SD = 3.54), with a minimum value of 18.0 and a maximum value of 36.0 and it was characterized by moderate right-sided skewness (skew = 0.52), with a slightly larger number of people who obtained lower scores.

#### 3.2.3. Resilient Coping

The mean score obtained on the BRC scale was 14.17 (SD = 2.74), with a minimum value of 8.0 and a maximum value of 20.0. The score obtained from the BRC was characterized by moderate left-sided skewness (skew = −0.13), which means a slightly larger number of people with higher scores.

### 3.3. Group Characterization

The participants were divided into two groups: G1 (N = 98), characterized by a lower level of anxiety with higher self-efficacy and resilient coping, and G2 (N = 47), with a higher level of anxiety with poorer self-efficacy and resilient coping. The profiles of both groups are given in Table 2. The characteristics of the groups are presented in Appendix A.

### 3.4. Willingness to Get a COVID-19 Vaccine

The vast majority of study participants had already been vaccinated against COVID-19 at the time of the study (N = 106, 73.1%), most often at a workplace vaccination site (N = 92, 86.79%). The analysis of potential correlations between psychological pattern and decision to vaccinate did not reveal a statistically significant correlation. In addition, the choice of vaccination site was not dependent on the group (Table 3).

A total of 12 respondents stated that they wished to be vaccinated in the near future, accounting for 32.4% of non-vaccinated participants. No decision on this subject was made by 14 respondents (37.8%). The groups were not significantly different in this regard either (Table 4).

### 3.5. Motivation to Vaccinate

The respondents did not differ in terms of the reasons for COVID-19 vaccination between the two groups (Table 5). Both groups were highly motivated related to the desire to protect their family and loved ones and, to a lesser extent, concern for their health.

### 3.6. Vaccine Information Sources

Regarding the various sources where respondents obtained information on vaccinations, only one was associated with a different frequency of vaccination depending on the group (Table 6). On average, respondents in G1 used nonspecialist blogs and vlogs less frequently compared to G2. A similar trend was observed regarding the use of non-health-related websites as sources of information on COVID-19 and vaccination.

### 3.7. Perception of the Vaccine and Vaccination

The respondents expressed their opinions on the causes of low vaccination rates in Poland. Significant intergroup differences were observed regarding the perception of causes. The comparison between G1 and G2 showed that respondents in G1 less commonly indicated reasons such as lack of faith in the effectiveness of the vaccine, lack of faith in the COVID-19 epidemic and restriction of civil liberty (Table 7).

## 4. Discussion

The vast majority of Polish novice nurses (PNNs) in the study group were vaccinated against COVID-19 and the analysis of levels of anxiety, self-efficacy, and resilient coping revealed that the levels were significantly different, which allowed us to distinguish two subgroups of the studied PNNs, G1 and G2.

One in four people surveyed did not vaccinate at all. In the group with a high level of anxiety, the proportion was one in five, and in the group with a low level of anxiety, it was three in ten. Within that group, only one in three said that they would not vaccinate in the future. Such a broad spectrum of attitudes related to COVID-19 vaccination is related to differences in the nature of individuals’ anxiety. Sometimes the experience can be highly emotional and lead people to make irrational decisions [39]. Sometimes it plays a functional role, instigating a change in behavior to adapt to negative stimuli [40]. Even if there are fewer unvaccinated people with higher levels of anxiety than those with lower levels of anxiety, we should not infer from this that anxiety in the population should be amplified in order to achieve higher vaccination rates [41]. Excessive anxiety and tension regarding the COVID-19 pandemic and vaccines could lead to impaired adaptive and preventive capability, leading to an increased number of unvaccinated individuals in the population [42]. The respondents indicated that they were more motivated to receive the COVID-19 vaccine to protect loved ones than to protect themselves. This attitude may be due to the widespread belief, which is confirmed in clinical practice, that COVID-19 affects elderly people much more severely than younger people [43]. Considering the fact that 84.1% of the respondents did not live alone (taking into account the demographics of the Polish society, it may be assumed that the vast majority of them lived with their parents), and every twelfth person (8.3%) lived with seniors, it is reasonable to consider vaccination with a view to protecting others more than oneself. 

Significantly, students who reported lower levels of anxiety were more likely to have relied on verified information posted on professional medical websites when seeking information about COVID-19 vaccinations. We build our beliefs based on the knowledge we obtain from various sources, and beliefs influence attitudes toward vaccine acceptance [44]. Such an observation has important implications in the context of the misinformation and anti-vaccine efforts that became apparent after the development and marketing of COVID-19 vaccines [45]. Studies conducted so far demonstrate that attitudes can be categorized and grouped [46]. Regrettably, surveying people directly with questionnaires is difficult, because those who display anti-vaccine attitudes constitute a minority that is viewed negatively by the government and the public. Therefore, some individuals may engage in self-censorship [47,48]. In this situation, people may develop the defensive behavior of complementary projection, whereby they assume that others share their beliefs, opinions, and priorities [49], and thus project their implicit beliefs onto them. Our study shows that the group with a higher level of anxiety and poorer self-efficacy was more likely to believe that other people did not get vaccinated for their own subjective reasons (right to choose, lack of faith in the efficacy of vaccination, or denial of disease risk). This is consistent with previous studies that linked anxiety with skepticism [50] and resistance to change [51]. 

### 4.1. Assumptions and Guidelines for Future Study

It is also worth noting the psychological characteristics of the study group participants in terms of anxiety, resilient coping, and self-efficacy. High levels of resilient coping and self-efficacy indicate the adaptive role of these two variables in new stressful situations. Nurses characterized by high levels of these two psychological traits had lower levels of anxiety compared to nurses who showed low levels of these traits. Low levels of anxiety and stress allow people to take more effective adaptive action. Nurses with low stress levels can recover faster in terms of psychological balance and act appropriately for the situation. It is justified to undertake activities aimed at shaping and strengthening resilience and self-efficacy in the context of coping with difficult stressful situations. Moreover, doing so may serve as a protective factor against burnout and job resignation. However, this requires further empirical research.

Despite statistically significant differences in terms of the severity of anxiety, the percentage of vaccinated nurses was the same in both groups. This suggests the existence of other psychosocial factors that were correlated with the decision to get vaccinated, which were not considered in this study. The severity of anxiety before receiving the vaccine is also unknown. The possibility of receiving the vaccine may have lowered anxiety in nurses with high levels of resilient coping and self-efficacy. However, such a mechanism would not be observed in the case of nurses with low resilient coping and self-efficacy. This is a theoretical consideration that requires further analysis. Therefore, the absence of recorded differences should be interpreted with great caution. A strong sense of responsibility for the health of the community in which one functions may constitute another important explanation for the lack of such differences. The high sense of responsibility for the health of others in the analyzed group of nurses is worth noting. Although this variable was not analyzed directly in the study group, expressions of the variable may be indirectly observed. For example, in both groups, regardless of the level of anxiety, resilient coping, or self-efficacy, there was strong motivation to vaccinate related to the wish to protect one’s family and loved ones and, to a lesser extent, concern for one’s own health. This may indicate the key role of this psychological variable in vaccine-related decision-making. Further research should include this variable. The sense of responsibility for one’s own health and that of the community in which one functions may be so strong that it motivates people to get vaccinated despite perceived fears.

In light of the data presented here and on the basis of existing evidence, it is extremely important to analyze the levels of anxiety in the group of novice nurses. Moreover, many novice nurses may consider reducing anxiety highly significant in terms of the desire to remain in the profession in the future

### 4.2. Limitations

The small size of the group of novice nurses is an important factor limiting the results of the present study. It is estimated that every year in Poland, about 12,000 students complete their studies in nursing [5], but only about 5500 are given the right to practice by the authorized body, the Chamber of Nurses and Midwives [52]. It is also known that a large number of graduates do not enter the profession [53]. Therefore, for objective reasons, it is impossible to estimate the actual number of novice nurses entering the profession each year. In addition, as already mentioned, the present study included nurses continuing their education toward a Master’s degree at medical universities in Poland. This group represents approximately 0.01% of all Master’s degree level nursing students in Poland (N = 2219) [54]. Due to objective difficulties in accessing professional groups of nurses and conducting the study with a large group during the COVID-19 pandemic, the authors decided to include novice nurses working in the profession and continuing their education at the master’s level.

It is worth emphasizing that the study participants comprised a special group of nursing graduates who had initially been taking classes online, and then in a hybrid form since March 2020. The pandemic situation, as in other countries worldwide, put into question the form and timing of their graduation and the possibility of entering the profession, which may have been an additional factor contributing to anxiety. 

## 5. Conclusions

Psychological variables may correlate with motivation, attitudes toward vaccination, and the choice of reliable sources of information about vaccination. Our study demonstrates the key role of two psychological variables, self-efficacy and resilient coping, in this context. Furthermore, these characteristics affect the ability to adapt to new working conditions among the group of novice nurses.

Professional adaptation is particularly important for nurses starting work in the profession, in the event that implementing crisis management of health protection systems, such as during a pandemic outbreak, becomes necessary. Such adaptation should particularly include an analysis of the levels of anxiety and a search for factors that may affect reducing it. This is a very important factor in ensuring adequate medical staff and maintaining the proper number of nurses and in protecting patient safety and the stability of health care systems.

## Figures and Tables

**Table 1 ijerph-19-15787-t001:** Characteristics of the study group (N = 145).

Medical University, N (%)	
Pomeranian Medical University	32 (22.1)
Poznan University of Medical Sciences	21 (14.5)
Wroclaw Medical University	19 (13.1)
Jagiellonian University Medical College	18 (12.4)
Medical University of Lublin	17 (11.7)
Medical University of Białystok	17 (11.7)
Medical University of Warsaw	12 (8.3)
Medical University of Gdańsk	9 (6.2)
Gender, N (%)	
Female	141 (97.2)
Male	4 (2.8)
Age (years)	
M ± SD	23.3 ± 0.92
Range	22.0–25.0
Working in hospital with COVID-19 units, N (%)	
Yes	46 (31.7)
No	99 (68.3)
Residence, N (%)	
Alone	27 (18.6)
With relatives/family/friends (excluding seniors)	106 (73.1)
With relatives/family/friends (including seniors)	12 (8.3)
Living with someone at higher risk of COVID-19, N (%)	
yes	24 (16.6)
no	121 (83.4)
COVID-19 infection from people in immediate environment, N (%)	
Yes, acute or very acute	37 (25.5)
Yes, but rather mild	78 (53.8)
No	25 (17.2)
I do not know	5 (3.4)
COVID-19 infection, N (%)	
Yes (acute symptoms of infection)	6 (4.1)
Yes (mild symptoms of infection)	34 (23.4)
Yes (no symptoms of infection)	5 (3.4)
Probably (no test confirmation)	18 (12.4)
No	61 (42.1)
I do not know	21 (14.5)

M, mean; SD, standard deviation.

**Table 2 ijerph-19-15787-t002:** G1 and G2 group profiles.

	G1 (N = 98)		G2 (N = 47)		t_df=143_	*p*-Value *	d ** (95% CI)
	M	SD	M	SD			
Anxiety	3.57	2.32	12.48	2.94	−23.538	<0.001	3.51 (2.98; 4.05)
Self-efficacy	27.07	3.30	25.22	4.51	3.332	0.001	−0.50 (−0.85; −0.14)
Resilient coping	14.42	2.39	13.09	2.98	3.430	0.001	−0.51 (−0.87; −0.16)

M, mean; SD, standard deviation; CI, confidence interval. G1: low level of anxiety, high self-efficacy and resilient coping; G2: high level of anxiety, poor self-efficacy and resilient coping. * Student’s *t*-test. ** Cohen’s d effect size.

**Table 3 ijerph-19-15787-t003:** Willingness to get COVID-19 vaccine in groups G1 and G2.

	Group 1		Group 2		*p*-Value *	OR (95% CI)
	N	%	N	%		
Vaccination against COVID-19						
No	30	30.61	9	19.15	0.166	1.86 (0.80; 4.33)
Yes	68	69.39	38	80.85		
Selected vaccination site						
Workplace	59	86.76	33	86.84	1.000	0.99 (0.31; 3.21)
University	9	13.24	5	13.16		

OR, odds ratio; CI, confidence interval. Group 1: low level of anxiety, high self-efficacy and resilient coping; Group 2, high level of anxiety, poor self-efficacy and resilient coping. * Fisher’s exact test (two-sided).

**Table 4 ijerph-19-15787-t004:** Willingness to receive COVID-19 vaccine in the future among unvaccinated respondents.

	Group 1		Group 2		χ^2^_df=2_	*p*-Value *
	N	%	N	%		
No	9	31.03	2	25.00	1.493	0.474
I do not know	12	41.38	2	25.00		
Yes	8	27.59	4	50.00		

Group 1: low level of anxiety, high self-efficacy and resilient coping; Group 2: high level of anxiety, poor self-efficacy and resilient coping. * Pearson’s chi-squared test.

**Table 5 ijerph-19-15787-t005:** Motivation to vaccinate in groups G1 and G2.

	G1 (N = 68)		G2 (N = 38)		t_df=104_	*p*-Value *
	M	SD	M	SD		
Concern about one’s health	4.53	1.51	4.08	2.17	1.253	0.213
Concern about family’s health	5.19	1.31	5.08	1.58	0.392	0.695

M, mean; SD, standard deviation. G1: low level of anxiety, high self-efficacy and resilient coping; G2, high level of anxiety, poor self-efficacy and resilient coping. * Student’s *t*-test.

**Table 6 ijerph-19-15787-t006:** Frequency of using sources of information on vaccination.

	G1 (N = 98)		G2 (N = 47)		t_df=143_	*p*-Value *
	M	SD	M	SD		
Websites of institutions and organizations related to health protection	3.96	1.61	3.95	1.60	0.017	0.986
Non-health-related websites	1.77	1.69	2.23	1.79	−1.795	0.074
Specialized professional journals (online and in print)	2.94	1.94	3.17	2.04	−0.786	0.433
Social media	2.38	1.73	2.60	1.89	−0.807	0.420
Blogs and vlogs of health care specialists	2.70	1.92	3.00	1.85	−1.058	0.291
Nonspecialist blogs and vlogs	0.62	1.07	1.02	1.27	−2.309	0.022
Web portals (e.g., Onet, Wirtualna Polska, etc.)	1.34	1.42	1.43	1.65	−0.393	0.695
Radio/television	1.73	1.50	2.11	1.98	−1.529	0.128
Classes at university	3.21	1.64	3.05	2.10	0.610	0.543
Workplace	3.31	2.26	3.65	2.29	−0.998	0.320
Other students at university	2.38	1.57	2.63	1.79	−1.031	0.304
Colleagues	2.84	1.95	3.08	2.10	−0.795	0.428
Family/friends not related to health protection	1.40	1.39	1.66	1.67	−1.186	0.237

M, mean; SD, standard deviation. G1: low level of anxiety, high self-efficacy and resilient coping; G2: high level of anxiety, poor self-efficacy and resilient coping. * Student’s *t*-test.

**Table 7 ijerph-19-15787-t007:** Perception of causes of non-vaccination in Polish society.

	G1 (N = 98)		G2 (N = 47)		t_df=143_	*p*-Value *
	M	SD	M	SD		
a. Confidence in government regarding vaccination	4.84	1.52	4.88	1.69	−0.160	0.873
b. Lack of faith in effectiveness of vaccine	4.68	1.40	5.29	1.11	−3.111	0.002
c. Fear of vaccine ingredients	4.97	1.33	5.17	1.22	−1.053	0.293
d. Fear of vaccine side effects	5.22	1.16	5.32	1.08	−0.625	0.533
e. Concern that vaccines were manufactured too quickly	5.17	1.27	5.11	1.34	0.348	0.728
f. Concern that vaccines have not been tested well enough	5.33	1.02	5.35	1.08	−0.162	0.871
g. Lack of faith in COVID-19 epidemic	3.69	1.68	4.37	1.77	−2.670	0.008
h. Restriction of civil liberty	3.50	1.78	4.15	1.61	−2.541	0.012
i. Belief that COVID-19 is not a serious disease	3.77	1.60	4.14	1.76	−1.498	0.136

M, mean; SD, standard deviation. G1: low level of anxiety, high self-efficacy and resilient coping; G2: high level of anxiety, poor self-efficacy and resilient coping. * Student’s *t*-test.

## Data Availability

Data available on request due to restrictions (privacy and ethical). The data presented in this study are available on request from the corresponding author.

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
