# Peer review of "How Psychological Variables Maybe Correlated with Willingness to Get COVID-19 Vaccine: A Nationwide Cross-Sectional Study of Polish Novice Nurses"

_ijerph, 2022, doi:10.3390/ijerph192315787_

Round 1

Reviewer 1 Report (Previous Reviewer 4)

The authors have addressed my concerns. I believe the manuscript is now suitable for publication.

Author Response

Thank you very much for positive conclusion and all of your comments and concerns.

Reviewer 2 Report (New Reviewer)

The authors of this article has examined the role of psychologic factors on nurses and their attitude on covid19 vaccines. This is a well written study that practicaly reveals the high anxiety levels of all the covid-hospital workers in the era of covid. Even though this is a subject already examined by other eg

1)Çelmeçe, Nuriye, and Mustafa Menekay. "The effect of stress, anxiety and burnout levels of healthcare professionals caring for COVID-19 patients on their quality of life." Frontiers in psychology 11 (2020): 597624.

2)Ofei-Dodoo, Samuel, Colleen Loo-Gross, and Rick Kellerman. "Burnout, depression, anxiety, and stress among family physicians in Kansas responding to the COVID-19 pandemic." The Journal of the American Board of Family Medicine 34.3 (2021): 522-530.

and many other studies, it is one of the little studies that aim on a specific group of workers and not the whole health care professionals. I suggest that this article deserves publication.

Author Response

Thank you very much for positive conclusion.

Reviewer 3 Report (New Reviewer)

The aim of the paper was to investigate whether there are any differences between two groups of novice nurses (a group with low levels of anxiety, high self-efficacy, and high resilient coping vs. a group with high levels of anxiety, poor self-efficacy, and low resilient coping) on the following variables: decision to vaccinate, reasons for COVID-19 vaccination, vaccine information sources, perception on the causes of low vaccination rate in the general population. The study tries to tackle an important topic: vaccination behaviour and motives in health personnel during a pandemic. Nurses were the largest medical group exposed to direct contact with the virus, hence vaccination was important for the health of the patients they were attending to, as well as for their own health.

-One major issue that I believe should be addressed is the lack of a clear argument for the strategy employed for data analysis. I am specifically referring to the creation of the two groups, which lay at the basis of all subsequent analyses. The authors describe that the k-means method was employed to distinguish two clusters, based on three indices: anxiety, self-efficacy, and resilient coping; the nurses that participated in the study were divided in two groups that supposedly reflect different psychological profiles. Looking into the literature in Psychology, we find that the three indices (anxiety, self-efficacy, and resilient coping) correlate with each other. I am assuming this might be the implicit argument that the authors had when choosing the three indices as basis for the clustering procedure. The authors need to state clearly and explicitly what is the theoretical (and/or empirical) argument(s) for expecting that the three indices will serve as a good basis for the creation of the two groups. Do the two different psychological profiles obtained via the k-means method reflect some sort of difference in general vulnerability? (e.g., would a person that has a high level of anxiety, poor self-efficacy, and a low level of resilient coping be more vulnerable, more prone to maladaptive behaviours?). Furthermore, one might think that repeating the same process of cluster creation on a different set of data might yield different results. Thus, it would be of the essence to have a set of arguments that also support the validity of this strategy. Otherwise, a different technique should be employed (e.g., multivariate multiple regression).

Introduction:

-While the authors offer many details regarding the status of nurses in general in the health system, the Introduction is missing key information on the variables investigated in the study. For instance, there is no mention on motivations toward COVID-19 vaccines, no information on vaccine information sources, no information on perception of the reasons for non-vaccination (why are you measuring these? why are they relevant? It should be stated explicitly).

-Although the authors speak of attitudes toward vaccination, I can’t seem to find any measure of the nurse's attitudes towards vaccination (only on motivations toward COVID-19 vaccines, which is not the same thing). The term attitudes also appears in the title and I believe it would be misleading.

-The authors state that “In light of the data presented here, it is extremely important to analyse the levels of anxiety and factors that influence its occurrence as well as ways to reduce anxiety among novice nurses in crisis situations such as the COVID-19 pandemic” – the results do not seem to show significant consequences of high anxiety (the only differences between G1 & G2 are that G2 -high anxiety et al.- uses Nonspecialist blogs and vlogs more frequently to learn about vaccination, and G2 make stronger assumptions about 3 of the 9 potential causes of non-vaccination in Polish society). If the authors decide to keep this idea (which should belong to the Discussion section), they should link it to specific results.

-After reading the Introduction, it’s not clear what are the hypotheses. The hypotheses should be clearly stated, either embedded in the Introduction, or at the end of this section.

As a general comment, the Introduction needs a complete restructuring, such that the information presented should lead to a logical formulation of the study hypotheses.

Methods:

-The following measures/questionnaires are not described: motivations and attitudes toward COVID-19 vaccines, vaccine information sources, perception of the reasons for non-vaccination. Although the items for each of these measures can be viewed in Tables 5, 6, and 7, the way participants respond to the items is not described.

-Also, if a questionnaire was developed for this specific study, the process of questionnaire creation should be described in detail.

Discussion:

- The authors state that “… being vaccinated did not significantly influence the analysed level of anxiety.” – this idea was not tested anywhere in the study. To state this, the authors would have to compare the vaccinated participants with the non-vaccinated participants regarding anxiety.

- The authors again refer to attitudes toward vaccination- is this is a direction for a future study? If so, it should be stated explicitly.

- The authors state that “…only one in three said that they would not vaccinate in the future. Such a broad spectrum of attitudes related to COVID-19 vaccination is related to differences in the nature of individuals' anxiety”. It seems that the authors conflate attitudes with intention to vaccinate. Also, the assumption that individuals' anxiety is related to differences in attitudes related to COVID-19 vaccination was not tested. If the authors did not directly verify a statement they make, it should be made clear they are only making an assumption, and this assumption could be tested in future research.

The Discussion section would need a major restructuring. The reader needs to be able to distinguish between statements that the authors have made based on the results vs. assumptions (new hypotheses that the authors derive on the basis of the results, that could be tested in future research).

Generally, the term “influence” (which also appears in the title and several times throughout the paper) is a strong term when it comes to research. Strong, in the sense that it implies a causal relation. Ideally, to investigate causal relations, we would need an experimental design, or at least a longitudinal design. Although some theories state that personality traits/psychological variables lead to specific outcomes, when employing cross-sectional design (as is the case here), formulas like X influences Y should be avoided.

Author Response

This manuscript is a resubmission of an earlier submission. The following is a list of the peer review reports and author responses from that submission.

Round 1

Reviewer 2 Report

I read the paper by Gotlib and colleagues carefully.

Although the topic, although already analysed extensively in the literature, is potentially interesting by providing the view from the perspective of Poland, I must say that I was extremely disappointed by the quality of the paper.

First of all, it is written in low-quality English, which often makes it unintelligible. It becomes difficult to give examples because the entire paper is riddled with macroscopic errors.

As for the content and not the style, this paper conducts even quite sophisticated analyses on a too small sample (just 149 nurses) about whom we do not know, just to mention a few major flaws:

1) how they were collected?

2) was there an ethics committee?

3) was there a potential comparison group?

The tables are also unclear, the statistical analysis discreetly conducted but not very intelligible.

I must, despite myself, say that this paper cannot be published and there is no margin to make it acceptable for publication in IJERPH, in my opinion.

Reviewer 3 Report

Thanks for the chance to review this manuscript. Please see below some points, all minor:

1- Results in Abstract does not reflect the title. Please shorten the Background and Methods and add more results

2- Only third of your participants who have not been vaccinated. How this may affect your results? please discuss this point.

3- In Table 2, please clarify what does (**) mean?

4- In the Discussion, the first two paragraphs are background not discussion. Discussion should start with a summary of the results. Please remove or integrate them in the Introduction.

5- Discuss results from other similar sectors like healthcare students. Please use this example (10.3390/vaccines9060665).

6- iThenticate report: the below paragraphs show high similarity or complete copy form published articles, please adjust them:

Lines 140-148

Lines 189-200

Lines 228-236

Lines 385-392

Reviewer 4 Report

I was pleased to read the manuscript by Gotlib et al. Overall, the study is well done and requires only minor language corrections. 

Before being suitable for publication, I think it is necessary for the Authors to better specify how the participants were selected.
The sample is very small and ultra-selected, as correctly addressed in the Limitations section. However, it would be necessary to better define the selection criteria ("coordinators selected the participants from a list of 122 graduates who, in their judgment, should take part in the study") which could be an additional source of biases. 

Round 2

Reviewer 2 Report

I am sorry to notice how the authors have not improved their paper sufficiently and also did not change the tables or other parts I suggest to them that were unintelligible.
